# An In-depth Investigation of User Response Simulation for Conversational Search

## Abstract

Conversational search has seen increased recent attention in both the IR and NLP communities. It seeks to clarify and solve users' search needs through multi-turn natural language interactions. However, most existing systems are trained and demonstrated with recorded or artificial conversation logs. Eventually, conversational search systems should be trained, evaluated, and deployed in an open-ended setting with unseen conversation trajectories. A key challenge is that training and evaluating such systems both require a human-in-the-loop, which is expensive and does not scale. One strategy is to simulate users, thereby reducing the scaling costs. However, current user simulators are either limited to only responding to yes-no questions from the conversational search system or unable to produce high-quality responses in general.

In this paper, we show that existing user simulation systems could be significantly improved by a smaller finetuned natural language generation model. However, rather than merely reporting it as the new state-of-the-art, we consider it a strong baseline and present an in-depth investigation of simulating user response for conversational search. Our goal is to supplement existing work with an insightful hand-analysis of unsolved challenges by the baseline and propose our solutions. The challenges we identified include (1) a blind spot that is difficult for the model to learn, and (2) a specific type of misevaluation in the standard empirical setup. We propose a new generation system to effectively cover the training blind spot and suggest a new evaluation setup to avoid misevaluation. Our proposed generation system leads to significant improvements over existing systems and large language models such as GPT-4. Additionally, our analysis provides insights into the nature of user simulation to facilitate future work.

## CCS Concepts

• **Information systems → Users and interactive retrieval**.

## Keywords

conversational search, user response simulation

## 1 Introduction

A study by Spink et al. [47] suggested that almost 60% of web search queries have fewer than three words. Conventional search systems usually perform a single-turn result retrieval based on a

*Conference'17, July 2017, Washington, DC, USA*
© 2023 Association for Computing Machinery.
ACM ISBN 978-x-xxxx-xxxx-x/YY/MM...$15.00
https://doi.org/10.1145/nnnnnnn.nnnnnnn

potentially ambiguous user query, implicitly placing the burden on users to go through the entire search result page to hopefully get the information they need. User behavior analysis and advanced natural language processing models have made it easier to clarify and iterate over complex user needs through conversations. This has led to the development of interactive information-seeking systems, usually referred to as *Conversational Search systems* [56]: an increasingly popular research topic and an essential frontier of IR [3, 12].

Most research about conversational search [1, 2, 5, 44, 45, 51, 54] is limited to training their system on datasets with observed or artificial conversation logs. Such a dataset would lack training signals and evaluation references when a conversation veers away from the dataset, especially when the system generates a question not listed in the dataset and steers the conversation in an unseen direction. We refer to this as the *open-ended* nature of a multi-turn conversational system, as opposed to training and evaluating the system with the recorded conversation trajectories from a dataset. Eventually, conversational search systems should be trained, evaluated, and deployed in an open-ended setting.

However, training and evaluating them in such a setting is challenging; it requires humans to generate responses to open clarifying questions, which can quickly get expensive and does not scale.

Past work [e.g., 40, 45] has demonstrated that a user response simulator that automatically generates human responses can help evaluate conversational search systems. Such a system aims to generate user-like answers to system-generated clarifying questions based on a query and the user's search intent. A user response simulation system can also enable studies such as training a multi-turn conversational search system by generating synthetic conversations and rewards and perhaps using Reinforcement Learning from Human Feedback (RLHF) [30].

The primary goal of this paper is to analyze and provide insights into the task of user response simulation for conversational search, focusing on the challenges with existing models and how the challenges may be addressed and identifying what is left to be solved. We conduct a manual analysis (Sec. 4) on a subset of a widely-used conversational search dataset Qulac [2]. We study all the low-scoring cases where a strong baseline model struggles. From the analysis, we conclude a categorization of its failings. Our analysis suggests that among all the low-scoring cases for our baseline model: (1) 10% of them contain out-of-context information either in the question or the reference response and therefore are extremely hard to be simulated by any system. (2) 38% of the generations are bad because the baseline model generates answers of the wrong type. (3) at least 45% of the generations are reasonable but misevaluated by the existing evaluation setup that ignores an important user variable, *cooperativeness*.

Then, we demonstrate a simple two-step generation system (Sec. 5.1, 5.2) that aims to address the answer typing errors in the abovementioned problem (2). Upon this, we are the first to

suggest that the task of user response simulation for conversational search generally resembles the task of question answering while also having distinguishing characteristics. We combine transferrable question-answering knowledge with an answer type predictor to jointly improve answer typing accuracy. In addition, we propose a new cooperativeness-aware evaluation heuristic to reduce misevaluations described in the abovementioned problem (3).

We test and compare our proposed system and heuristic with existing approaches and large language models (LLMs) in Sec. 6. The evaluation of them is done from four perspectives. The first is text generation metrics, such as BLEU [32], ROUGE [26], and METEOR [4]. Then, we evaluate answer type correctness using the classification F1. Next, we compare the document retrieval performances of search after appending the clarifying question and the generated response to the initial query, following [2, 40, 45]. Finally, we employ crowd workers to score the generated responses regarding generation relevance and naturalness.

Our experiment results (Sec. 7) show that our proposed system significantly outperforms existing approaches and LLMs. Human judgments confirm that our proposed system generates more relevant and natural responses. Our full experiment code and human annotation results are published on Anonymous GitHub [1].

## 2 Related Work

This section gives a high-level overview of previous work from two related fields: conversational search and user response simulation. Other related work is introduced in their appropriate context.

**Conversational Search** Conversational search is a novel search paradigm that aims to clarify and solve users' complex search needs through natural language conversations [56]. Recent work [e.g., 3, 12] has identified it as one of the research frontiers of IR, and it has been the focus of a large volume of seminars and surveys [e.g., 3, 16–19, 56]. The most desired feature of conversational search is that both the user and the system can take the conversational initiative as suggested in the theoretical framework by Radlinski and Craswell [36]. Zamani and Craswell [53] later proposed a conceptual pipeline of such a mixed-initiative conversational search system. In most existing conversational search systems [2, 39, 55], system-initiative is implemented as proactively asking clarifying questions about the search query. Evaluating these systems and scaling their functions to multi-turn systems require an actual human to provide feedback for their clarifying questions. Because human-in-the-loop is expensive and not scalable, evaluation and scaling remain challenging for conversational search systems.

**User Response Simulation for conversational systems** User Simulation for conversational systems has been broadly studied by NLP and IR communities in the past [15, 23, 42, 43]. One of the earlier user simulation methods is agenda-based simulation [41], where users are assumed to generate responses around a specific dialogue objective. It has been shown to be effective for various close-domain tasks such as task-oriented dialogue systems and conversational recommendation. In these tasks, a simple set of rules can usually lead to highly realistic systems [20, 25, 57].

---
[1] https://anonymous.4open.science/r/UserSimulation-7091

Some recent work [7, 8, 58] used deep learning and reinforcement learning to learn user simulation from data.

User simulation for open-domain conversational information retrieval is relatively under-explored [31, 40, 45]. Salle et al. [40] demonstrated a multi-turn search intent clarification process of conversational search with a clarifying question selection model and a user response generation model named CoSearcher. In their system, the user simulator needs to respond to clarifying questions in the form of whether the search intent is a guessed intent from the clarifying question selection model. Their model also considers various user parameters such as the user's *patience* for engaging in the conversation and *cooperativeness* for supplying the user response with more information about the search intent. However, their system is limited to only responding to 'yes-no' questions and selects a response from the dataset instead of generating a response. In this work, we propose to answer all types of clarifying questions; Section 3.2 presents our categorization of clarifying question types.

Sekulić et al. [45] proposed a GPT-2-based [35] generative user simulation model named USi, which can generate responses to any clarifying question for open-domain conversational search. Their model is shown to have human-like performance. Later, Owoicho et al. [31] exploited a GPT-3-based few-shot prompting approach to generate user answers to clarifying questions.

This paper shows that the similarity of user response simulation to question answering (QA) can improve the former with knowledge learned from QA, and better answer typing can bring further improvements. We also show that zero-shot large language models are inadequate for reliable user simulation. Further, we conduct an in-depth investigation, propose solutions, and provide insights into simulating user responses for conversational search.

## 3 User Response Simulation Task

This section defines the user response simulation task for conversational search and briefly introduces the datasets we use.

### 3.1 Task Definition

A conversational search session starts with the user issuing a potentially ambiguous search query to the search system. For example, a user may look for an anti-spyware program called *Defender* and type "Tell me about defender" in the search system. The word 'defender' is ambiguous: it can also refer to other concepts, such as a TV series, a vehicle model, or a video game. The conversational search system may want to clarify whether the user is looking for the TV series by "Are you interested in a television series?" Existing work and datasets [1, 2, 13] show that the user could respond in various ways, such as

No.

That is not related to my search.

Software.

No, I am looking for a software named Defender.

Despite their differences, all of them are consistent with the original search intent. Therefore, we define our user response simulation task as follows: Formally, given the user search query $q$, search intent $i$, and clarifying question $cq$, a user response simulation system should generate an answer $a$ in natural language that is consistent with $i$. Specifically, in the above example,

**Table 1: Clarifying question distributions by answer type.**

| Answer Types | Qulac/train | ClariQ/train |
|---|---|---|
| yes—confirming | 18.3% | 19.6% |
| no—negating | 57.9% | 54.6% |
| open-answer | 20.3% | 22.3% |
| irrelevant | 3.6% | 3.5% |
| Dataset Size # | 5273 | 8566 |

> $i$ = "I am looking for an anti-spyware program, Defender."
> $q$ = "Tell me about defender"
> $cq$ = "Are you interested in a television series?"
> $a$ = "No, I am looking for a software named Defender."

## 3.2 Datasets and Challenges

User simulation is still an underexplored research area without many sizable datasets. We use two publicly available datasets in this work: Qulac [2], and ClariQ [1] for easier comparisons with previous work about user simulation [40, 45]. Qulac dataset is built based on faceted queries from TREC Web Track (Clueweb) 09-12 [11]. It contains rows of queries, facets, clarifying questions, and answers representing one turn of conversational search, where the clarifying questions and answers are generated by crowd workers. The dataset can be seen as tree-structured, where each faceted query has multiple facets and multiple reasonable clarifying questions. We use the Qulac training set for finetuning our systems, the development set for studying the problem, and the test set for evaluation. ClariQ extends Qulac with additional queries and facets and creates a new test set with topics not included in Clueweb09-12.

***Multiple Clarifying Types*** As we previously mentioned, the CoSearcher [40] system only simulates a specific type of clarifying questions that can be answered by 'yes' or 'no' (also referred to as check [24] or verification [21] questions). However, we notice that many clarifying questions in their datasets cannot be answered by 'yes' or 'no'. We find that questions not answerable by 'yes' or 'no' are mostly open questions or Wh-questions. In addition, we notice many questions with answers expressing uncertainty or irrelevancy, such as "I don't know." or "This is not related to my search.", etc. This category is necessary to respond to questions to indicate that it is irrelevant and does not actually answer the question. Therefore, we extend the heuristic rule used in CoSearcher and categorize all the questions by their answer types into four classes: {yes, no, open, irrelevant}, as we show their distribution in Table 1. Our extended rules are shown in Alg. 1 in the appendix:

***Unknown User Cooperativeness*** Whether a user provides extra information besides minimally answering the clarifying question with 'yes' or 'no' is defined as *cooperativeness* in [40]. In the example in Sec. 3.1, the 1st and 2nd responses are uncooperative, while the 3rd and 4th are cooperative. After we inspect the two datasets, we find that crowd-worker-generated responses seem to have random cooperativeness, even when they have the same search intent or they are answering the same clarifying question, i.e., the cooperativeness is unpredictable given $(i, q, cq)$. Because of this, using all of the examples from the datasets indifferently to train a single system as USi [45] to generate both types of answers could be challenging.

## 4 T5: A Strong Baseline

We now introduce the T5 baseline and its failings in user simulation.

### 4.1 T5 Model

Our baseline for this task is to finetune a pretrained T5 [37] checkpoint on our training sets. T5 is a general-purpose text generation model pretrained on extensive text-to-text tasks, including summarization, machine translation, and question answering. Further, T5 is the strongest open-source model we can finetune. T5 has the same structure as a conventional encoder-decoder transformer [50], with simplified layer norm and relative positional embeddings. As a seq2seq generator, T5 is shown [37] to perform better than decoder-only models like GPT-2 [35] that was used in existing user simulator [45]. It first encodes the input text sequence and then generates an output text sequence with step-by-step decoding. The input and output sequence we present to T5 are formatted as follows:

$$input\_seq = i \; . \; q \; . \; cq$$
$$output\_seq = a$$

Here, $i$ is the user search intent, $q$ is the user query, $cq$ is the system-generated clarifying question, and $a$ is the answer to the question. During finetuning, we pass the input and output examples to pretrained T5 and tune it until convergence with cross-entropy loss.

Although the input and output of T5 and GPT-2 are almost identical, they are structurally different. With a complete encoder-decoder structure, T5 outperforms GPT-2, which only has a decoder. The comparison of T5, GPT-2, and zero-shot GPT-3.5 is shown in Table 2, where we see that finetuned T5 already outperforms current user simulation systems [31, 45]. Yet these scores seem far from being perfect. What is T5 missing for user simulation?

### 4.2 A Deep Dive into T5 Generations

To answer the above question, we conduct an in-depth investigation of the task by studying the cases of T5 with significantly low ROUGE scores. Our analysis is done on the Qulac dev set with the output from T5 finetuned on the Qulac train set. We intuitively keep all the examples with ROUGE lower than 0.2, representing generations that are unlike the human generations. This results in a subset of the Qulac development set with 360 generation examples, which is 27.9% of the development set. We investigate these examples and try to find out why the scores are so low. In doing so, we identify a few common types of low-scoring examples, described below.

***Type 1: Answering clarifying question requires extra information.*** This class comprises questions that ask for user-specific information such as address, age, preference, etc. For example, (In examples for this section, $G$ is the T5-generated responses, and $H$ is the human-generated response):

> $i$ = "Where can I find cheat codes for PlayStation 2 games?"
> $q$ = "PS 2 games"
> $cq$ = "What types of PS 2 games do you like to play?"
> $G$ = "I want to find cheat codes for PlayStation 2 games."
> $H$ = "Role playing."

In this example, there is no explicit information about the user's preferred game genre. Thus the system does its best - to answer with the user's true intent. These examples are hard to be simulated

**Table 2: T5-small (finetuned) outperforms USi (finetuned) [45] and ConvSim (zero-shot) [31]. † indicates $p < 0.01$ statistical significance of improvements over GPT-3.5—four orders of magnitude larger than T5-small—using permutation test [14, 46].**

| Dataset | Model | Generation Metrics | | | | Retrieval Metrics | | | | |
|---|---|---|---|---|---|---|---|---|---|---|
| | | BLEU-3 | BLEU-4 | ROUGE-L | METEOR | nDCG1 | nDCG5 | nDCG20 | P@1 | MRR |
| Qulac | GPT-2 (USi[45]) | 12.6 | 9.1 | 28.2 | 28.9 | 0.185 | 0.186 | 0.173 | 0.244 | 0.352 |
| | GPT-3.5 (ConvSim [31]) | 13.5 | 9.8 | 29.1 | 29.0 | 0.195 | 0.193 | 0.177 | 0.255 | 0.365 |
| | T5-small | 23.7† | 19.0† | 40.8† | 43.2† | 0.217† | 0.210† | 0.188† | 0.281† | 0.390† |
| ClariQ | GPT-2 (USi[45]) | 13.5 | 9.8 | 28.8 | 28.6 | 0.135 | 0.122 | 0.106 | 0.160 | 0.233 |
| | GPT-3.5 (ConvSim [31]) | 13.4 | 9.7 | 28.9 | 28.4 | 0.142 | 0.131 | 0.114 | 0.167 | 0.242 |
| | T5-small | 24.3† | 19.5† | 41.0† | 43.3† | 0.150† | 0.134† | 0.118† | 0.176† | 0.249† |

faithfully without additional information such as user profiles. We find 37 examples in this class, about 10.3% of the studied set.

**Type 2: Both generations are valid.** The examples in this class have equally valid T5-generated responses as human-generated ones. However, they are not correctly evaluated by the current automatic evaluation metrics because the word-overlap-based metrics cannot effectively evaluate paraphrases. For example:

> $i$ = "Find the homepage of the president of the United States."
> $q$ = "President of the United States"
> $cq$ = "Are you looking for a list of all US presidents?"
> $G$ = "No I want the homepage of the president."
> $H$ = "I need to go to his web site."

In this example, both the system and human-generated responses are valid. However, because they almost do not share words in common, the ROUGE score for the system generation is low. Among all the cases, 50 examples fall into this class, which is about 13.9%.

**Type 3: Cooperativeness mismatch.** As briefly mentioned in Sec. 3.2, *Cooperativeness* captures the following phenomenon: For yes-no questions, users tend to answer the question in various ways, with the difference in the amount of information. For example:

> $i$ = "How do I register to take the SAT exam?"
> $q$ = "SAT"
> $cq$ = "Do you need information about the San Antonio International Airport?"
> $G$ = "No I need to register to take the SAT exam."
> $H$ = "No."

Here, both the human-generated and T5-generated answers are valid. The human-generated response represents an uncooperative user who tends to answer with minimal effort. The T5-generated response contains more information and represents a cooperative user. Because the text generation metrics are sensitive to sentence lengths, the generation gets low scores. This reason applies to 112 examples among all the low-scoring cases, which is about 31.1% of the studied set.

**Type 4: Generating wrong answer type.** As mentioned in Alg. 1, we introduced a 4-way categorization of answer types. This class contains examples where the T5 generates the wrong answer type, e.g., when it needs to say 'yes'; instead, it says 'no' or vice versa. When this type of mistake happens in answering clarification questions, the response mostly has the wrong meaning. For example:

> $i$ = "I'm looking for web sites that do antique appraisals"
> $q$ = "Appraisals"
> $cq$ = "Do you need an antique appraised?"
> $G$ = "No I want to know about antique appraisals."
> $H$ = "Yes."

In this above example, the T5-generated response is a 'no'-type answer, which determines that the meaning of the answer is contrary to the human-generated answer. We find 122 examples of such mistakes, which is 33.9% of all cases.

**Type 5: Noise in data.** The clarifying questions or the human-generated responses can be of poor quality sometimes because they are crowd-sourced. A bad clarifying question does not clarify the search intent, and it can be challenging to generate a valid response to it. A bad human-generated response can be incoherent or inconsistent with the search intent and cause good generation getting low evaluation scores. For example:

> $i$ = "Find information on various types of computer memory, and how they are different."
> $q$ = "Memory"
> $cq$ = "Who was the first to study the brain and memory?"
> $G$ = "I want to know how different they are."
> $H$ = "Herman Ebbinguaus."

*Herman Ebbinguaus* is a pioneer of brain and memory studies. However, the $cq$ 'Who was ...?' is not a clarifying question for the query. A valid clarifying question could be, 'Do you want to know who was ...?'. The human-generated response is also unnatural, as the user is the information seeker, not the provider. Therefore it would be hard for T5 to generate any meaningful response. There are 21 examples in this class, which is about 5.8%.

**Type 6: Miscellaneous.** The rest of the examples are all in this class, where we find the T5 generations are wrong for various reasons but different from any of the abovementioned classes. Most of them are isolated, wrong generations for a plethora of reasons. There are 15 examples in this class, about 4.2% of the studied set.

## 4.3 A Summary of T5's Failings

Table 3 shows the distribution of low-scoring causes of T5, where the main reasons are generating the wrong answer type and cooperative mismatch. A few other observations are worth noting: (1) At least 45% of the low-scoring generations are good. This number is obtained by adding the 'Cooperativeness mismatch' and 'Both valid'

**Table 3: Categorization of reasons for low ROUGE**

| Reasons | T5 |
|---|---|
| Wrong answer type | 33.9% |
| Cooperativeness mismatch | 31.1% |
| Both valid | 13.9% |
| Extra information | 10.3% |
| Noisy reference | 5.8% |
| Miscellaneous | 4.2% |
| Total # ROUGE<0.2 | 360 |

cases. (2) Only 38% of the low-scoring generations are actually bad, by summing up the 'Wrong answer type' and 'Miscellaneous' types. This represents the actual spaces for improvement over T5. This analysis eventually tells us that the most realistic way to improve system performance is to address the answer type errors in the system and to avoid cooperative mismatch during evaluation. Our solutions to address these will be described next in Section 5.

## 5  QA-Enhanced User Simulation

This section describes our proposed models, which aim to address the failings of T5 and the misevaluation.

### 5.1  Pretraining from Question Answering

Simulating user responses to clarifying questions is similar to question answering (QA) tasks in NLP in that both require a response to a question given contexts (search intent).

Can we improve user simulators from knowledge of these tasks and QA tasks in general? We instantiate our experiments using one of the current state-of-the-art models for QA—UnifiedQA [22], which extends T5 by training on twenty QA datasets across four formats. One examplary dataset in UnifiedQA is BoolQ [10], which consists of yes-no questions with a short paragraph provided as context. The verification-type clarifying questions in user simulation, such as "Are you looking for X?", can be considered a particular case akin to the examples in BoolQ. As another examplary QA dataset in UnifiedQA's training set, SQuAD [38] contains reading comprehension questions with context. The wh-questions in SQuAD are also similar to the open-type clarifying questions in user simulation. Further, even questions that do not directly map to user simulation can potentially increase the general reasoning ability of the simulation system, according to the UnifiedQA paper.

We finetune the pretrained UnifiedQA on our dataset following the format instructions in UnifiedQA. We treat the intent $i$, query $q$ as the context, and $cq$ as the question. During training, we found that the query $q$ does not provide performance gain and can be dropped from the context. Therefore, our final input and output format for finetuning UnifiedQA is as follows:

$$input\_seq = cq ? \setminus n \text{ I am looking for } i$$
$$output\_seq = a$$

where '\n' is a unique backslash-n character, as advised in UnifiedQA. Adding 'I am looking for' is because most of the intents from the dataset (the facet column) are imperative sentences such as "Find information about human memory". Therefore, we add the prefix to mimic questions in the UnifiedQA training tasks. It can also be considered as a form of prompting [27, 33, 35].

## 5.2  Answer-Type-Driven User Simulation

From the analyses in Section. 4.2, we find that the most common error of the original T5 is generating wrong answer types. Naturally, the most important word in the answer to a yes-no question will be the 'yes' or 'no'; it almost solely determines the semantics and sentiment of the rest of the answer. Therefore, there should be a higher priority to correctly generate the 'yes' or 'no' over the other words in the response. However, we find that UnifiedQA is good at generating various possible answers but may not be good at predicting which answer type is correct.

Specifically, we find two incongruous cases of the top 10 beams from UnifiedQA when the search intent is "I am looking for X." and the clarifying question is "Are you looking for Y?". The first case contains both "Yes, I am looking for Y" and "No, I am looking for X.", simultaneously. The second case contains both "Yes, I am looking for X" and "No, I am looking for X".

Therefore, we can leave the answer typing task to a specialized model, such as a classification model. To this end, we propose to train a RoBERTa [28] classifier that predicts answer types to guide the UnifiedQA through constrained generation, as RoBERTa is a representative state-of-the-art text classifier. We fine-tune a pretrained RoBERTa-base model on the answer-type classification task, as defined in Table 1. During inference, we convert its prediction to generator decoding constraints as follows: If the predicted answer type is 'yes' or 'no', then the generation starts with 'yes' or 'no' correspondingly. If the predicted answer type is 'irrelevant', the generation starts with 'I don't know'. Otherwise, we will not place any constraints on the generator. We refer to this pipeline as Type+QA in our later sections.

## 6  Experiments

Our experiments aim to test whether our proposed models can effectively improve upon existing models and how well the challenges we identify in our analyses can be resolved. All the experiments are done on Qulac and ClariQ dataset.

### 6.1  Research Questions

***Q1: Can QA and answer typing help user simulation?*** Our first research question is whether QA knowledge and adding the extra step of answer type prediction can improve user simulation quality. To answer this question, we compare our proposed Type+QA model with the finetuned T5 and QA-enhanced baselines.

***Q2: Can cooperativeness-awareness evaluation reduce misevaluation?*** The analysis in Sec. 4 shows that a large proportion of mis-evaluation is due to the cooperativeness mismatch from the random cooperativeness challenge mentioned in Sec. 3.2. We propose that the datasets, if they are to be used for user response simulation, need a necessary column indicating the cooperativeness. However, this issue is rarely mentioned in existing work [40].

Therefore, we propose a cooperativeness-aware heuristic to train and evaluate user simulation systems. We partition the dataset into two subsets: one with short generations of fewer than three words as the uncooperative group and the rest as the cooperative group. Next, we train two simulation systems on each partition and evaluate them separately on each partition.

**Q3: How good are zero-shot LLMs for user simulation?** Large language models (LLMs) are models trained with numerous next-token-prediction tasks to learn general-purpose language representations for nearly any language understanding or generation task. Its applications, such as ChatGPT [29], are shown to exhibit human-level performance on various benchmarks, with tasks similar to the user simulation task. Therefore, we want to explore to what extent can LLMs be directly used for user simulation. Because of the enormous sizes of LLMs, we cannot fully download or train them. Therefore, all these experiments are done in a zero-shot setting.

Our experiments with LLMs are divided into two groups: (1) open models including Llama-2 [49] and Flan-T5 [52] and (2) commercial models including GPT-3.5 and GPT-4 [29]. Open models are significantly smaller and downloadable, which can facilitate reproducibility. Commercial models are larger and better-performing but are only accessible through APIs and unnecessarily reproducible.

Prompting is an important aspect of using these LLMs as it can change their behaviors completely. In the experiments, we strictly follow the exact same prompting formats from their paper. For GPTs, we empirically choose the best-performing prompting instructions, which are included in the appendix.

### 6.2 Evaluation

In this section, we investigate four evaluation paradigms and describe their limitations which are not mentioned by existing work.

**Text Generation Metrics** The text generation metrics measure the similarity between generated response and the human reference by word overlap, such as BLEU [32], ROUGE[26], and METEOR [4]. These overlap-based metrics are imperfect, as they are sensitive to paraphrasing. As a result, a system-generated response could have the same meaning as the human reference yet get low scores.

**Answer Type** As we have discussed in Sec 4.1, answer type is an important reason for simulation failings. However, it could not be properly measured by the above generation metrics, because answer type is usually affected by a few keywords such as 'yes' or 'no'. For example, against a human reference "No, that is not what I am looking for.", "No." is still semantically better than "Yes, that is what I am looking for." The generation metrics could give bad generations the exact opposite meaning higher scores than good generations in this example. Therefore, we propose to include classification F1 for answer type as an auxiliary metric. We use the Alg. 1 in the appendix for classifying generated responses.

**Retrieval Metrics** Evaluating the document retrieval performance using retrieval models like e.g., query-likelihood model [34] is a standard paradigm in existing work [2, 31, 40, 45]. This evaluation is based on the motivation and assumption of asking clarifying questions in conversational search: the additional information from the question and its responses should retrieve better results.

**Human Judgement** We hire crowd workers from Amazon MTurk to evaluate the generated responses. The workers are required to have at least 500 lifetime HITs approvals and 95+% approval rate. We provide workers 200 randomly sampled tuples of Qulac query, search intent, clarifying question, and generated response, with a shuffled list of generations from different models without knowing the source. We ask the workers to score the generated responses

according to two criteria, *relevance* and *naturalness*, from 1 to 5. The workers get paid at twelve dollars per hour.

*Relevance* is defined as whether the response is consistent with the search intent and whether it helps the search system better understand the user's unspecified intention. Relevant answers perfectly align with the intent, while irrelevant responses contradict the search intent or can be randomly off-topic. Similar existing metrics with different names have been seen in prior work, such as adequacy [48], informativeness [9], and usefulness [45].

*Naturalness* measures whether the generated response is fluent, grammatical, and human-like. In contrast, an unnatural answer might have logical errors in them, or perhaps be impossible to understand. Moreover, natural answers should not provide information beyond what the question asks for.

## 7 Results and Analyses

We present the evaluation results for the oracle models, zero-shot LLMs, and finetuned models including T5 and our proposed model in Table 4 and 5, which are meant to be comparable with Table. 3 in the Qulac paper [2], Table. 3 in the USi paper [45], and Table. 1 in Cosearcher paper [40]. These results show that Human evaluation results are shown in Table 6. Additional manual analyses in Table 7 show that our proposed system has significant reduced low-scoring generations of corresponding reasons.

**Observations from the Oracle Models.** We include three oracle models as baselines to provide insights for understanding the numbers in our tables. The 'Query-only' row does not generate a response, it shows the document retrieval performance of searching without any interactions but only with the query. Unsurprisingly, its result is always the worst. The 'Human' row is the human-generated response. Thus it has the perfect score for text generation metrics. The 'Copy-intent' row is an Oracle model that always copies the search intent as the user response.

The goal of the 'Copy-intent' model is to represent an unnatural baseline user simulator that only cares about leaking the true search intent to the search system. Its generation scores are noticeably low, showing that real humans tend to differ from simply repeating the search intent. We can see from Table 4 and 5, the copy-intent model consistently achieves better document retrieval performances than humans, suggesting that the document retrieval metric does not fully align with human likeness.

**Type+QA improves the T5 baseline in automatic evaluation.** From Table 4 and Table 5, Type+QA consistently outperforms the T5 baseline with statistical significance in most columns. In particular, the F1 scores of the Type+QA model are significantly higher than T5 and using UnifiedQA alone. This shows that the Type+QA model effectively predicts the correct answer type, addressing the most common error of the T5 baseline. Being the best in text generation metrics also suggests it produces the most human-like generations.

The only columns where Type+QA does not outperform T5 are nDCG1 and P@1. However, none of their performance differences in document retrieval are significant, as they only differ in the third decimal place. Both T5 and Type+QA document retrieval scores are higher than humans and on par with the copy-intent baseline, indicating that they have indistinguishable utilities for retrieval.

**Table 4: Our proposed Type+QA model outperforms the T5 model on Qulac dataset. Refer to Section 7 for detailed explanations. Bold numbers indicate the highest performance of the column excluding the Oracles. † indicates $p < 0.05$, and ‡ indicates $p < 0.01$ statistical significance of improvements over finetuned T5-small using permutation test [14, 46].**

| | Model | Type F1 | Generation Metrics | | | | Document Retrieval | | | | |
|---|---|---|---|---|---|---|---|---|---|---|---|
| | | | BLEU-3 | BLEU-4 | ROUGE-L | METEOR | nDCG1 | nDCG5 | nDCG20 | P@1 | MRR |
| Oracles | Query-only | - | - | - | - | - | 0.133 | 0.146 | 0.153 | 0.190 | 0.294 |
| | Human | 100.0 | 100.0 | 100.0 | 100.0 | 93.7 | 0.198 | 0.195 | 0.178 | 0.259 | 0.367 |
| | Copy-intent | 8.3 | 17.3 | 13.8 | 31.4 | 29.4 | 0.216 | 0.212 | 0.193 | 0.283 | 0.391 |
| zero-shot | GPT-3.5 (ConvSim [31]) | 42.1 | 13.5 | 9.8 | 29.1 | 29.0 | 0.195 | 0.193 | 0.177 | 0.255 | 0.365 |
| | GPT-4 | **45.6** | 11.6 | 7.9 | 28.6 | 28.6 | 0.210 | 0.202 | 0.183 | 0.270 | 0.376 |
| | Llama2 | 23.6 | 6.5 | 4.5 | 19.9 | 18.6 | 0.189 | 0.187 | 0.172 | 0.248 | 0.350 |
| | Flan-xxl | 43.3 | 0.2 | 0.1 | 21.9 | 9.7 | 0.167 | 0.172 | 0.162 | 0.223 | 0.328 |
| finetuned | GPT-2 (USi[45]) | 24.4 | 12.6 | 9.1 | 28.2 | 28.9 | 0.185 | 0.186 | 0.173 | 0.244 | 0.352 |
| | T5-small | 34.1 | 23.7 | 19.0 | 40.8 | 43.2 | **0.215** | 0.209 | 0.188 | **0.279** | 0.388 |
| | QA-Enhanced | 41.7 | 23.3 | 18.7 | 40.9 | 43.4 | **0.215** | **0.210** | 0.188 | **0.279** | 0.390 |
| | Type+QA | 43.3$^{\ddagger}$ | **24.4**$^{\ddagger}$ | **19.6**$^{\dagger}$ | **41.6**$^{\ddagger}$ | **43.5** | 0.214 | **0.210** | **0.189** | 0.277 | **0.390** |

**Table 5: Our proposed Type+QA model outperforms the T5 model on ClariQ dataset. Refer to Section 7 for detailed explanations. Bold numbers indicate highest performance of the column excluding the Oracles. † indicates $p < 0.05$, and ‡ indicates $p < 0.01$ statistical significance of improvements over finetuned T5-small using permutation test [14, 46].**

| | Model | Type F1 | Generation Metrics | | | | Document Retrieval | | | | |
|---|---|---|---|---|---|---|---|---|---|---|---|
| | | | BLEU-3 | BLEU-4 | ROUGE-L | METEOR | nDCG1 | nDCG5 | nDCG20 | P@1 | MRR |
| Oracles | Query-only | - | - | - | - | - | 0.118 | 0.109 | 0.089 | 0.132 | 0.202 |
| | Human | 100.0 | 100.0 | 100.0 | 100.0 | 93.4 | 0.139 | 0.127 | 0.112 | 0.163 | 0.238 |
| | Copy-intent | 8.71 | 18.1 | 14.4 | 31.7 | 29.9 | 0.149 | 0.137 | 0.121 | 0.175 | 0.250 |
| zero-shot | GPT-3.5 (ConvSim[31]) | 41.9 | 13.4 | 9.7 | 28.9 | 28.4 | 0.142 | 0.131 | 0.114 | 0.167 | 0.242 |
| | GPT-4 | 45.5 | 10.5 | 7.3 | 28.7 | 34.8 | 0.146 | 0.134 | 0.117 | 0.170 | 0.245 |
| | Llama2 | 22.6 | 6.0 | 4.0 | 19.4 | 17.9 | 0.138 | 0.129 | 0.111 | 0.162 | 0.236 |
| | Flan-xxl | 44.4 | 0.2 | 0.1 | 21.4 | 9.5 | 0.132 | 0.121 | 0.104 | 0.154 | 0.227 |
| finetuned | GPT-2 (USi[45]) | 22.8 | 13.5 | 9.8 | 28.8 | 28.6 | 0.135 | 0.122 | 0.106 | 0.160 | 0.233 |
| | T5-small | 36.6 | 24.3 | 19.5 | 41.0 | 43.3 | **0.150** | 0.134 | 0.118 | **0.176** | **0.249** |
| | QA-Enhanced | 45.9 | 24.3 | 19.4 | 41.6 | **43.6** | 0.148 | 0.135 | **0.119** | 0.170 | 0.247 |
| | Type+QA | **46.3**$^{\ddagger}$ | **25.2**$^{\dagger}$ | **20.2**$^{\dagger}$ | **42.1**$^{\ddagger}$ | 43.1 | 0.149 | **0.136** | **0.119** | 0.172 | **0.249** |

**Table 6: Our proposed Type+QA mode outperforms T5 in crowd-source evaluation in both relevance and naturalness. ★ indicates $p < 0.01$ statistical significance of improvements over 'copy-intent', while † indicates $p < 0.05$ statistical significance over T5 using paired t-test.**

| model | Relevance | Naturalness |
|---|---|---|
| T5-small | 4.21 | 4.16$^{\star}$ |
| Type+QA | 4.35$^{\dagger}$ | 4.30$^{\star\dagger}$ |
| Copy-intent | 4.64 | 3.57 |

***Human evaluation confirms the improvements.*** Crowd-sourced human evaluation from Table 6 shows that the copy-intent oracle generates the most relevant responses while being poor in naturalness. This is strong evidence for our claim in Sec. 6.2 that document retrieval is a biased metric as it could favor unnatural generations. T5 simulator has higher naturalness over copy-intent, but lower relevance. Our proposed Type+QA model further improves both

**Table 7: Type distribution for low ROUGE generations. Our proposed Type+QA and Cooperativeness-aware evaluation effectively address T5's main type of failings—wrong answer type and cooperativeness missmatch, respectively. † indicates $p < 0.05$, and ‡ indicates $p < 0.01$ statistical significance of improvement over T5 using permutation test.**

| Reasons | T5 w/ Coop | T5 | Type+QA |
|---|---|---|---|
| Wrong answer type | 46.4% | 33.9%$\rightarrow$ | 11.8%$^{\ddagger}$ |
| Cooperativeness miss | 2.4%$^{\ddagger}$ | $\leftarrow$ 31.1% | 44.6% |
| Both valid | 16.4% | 13.9% | 15.2% |
| Extra information | 15.6% | 10.3% | 13.0% |
| Noisy reference | 11.6% | 5.8% | 7.4% |
| Miscellaneous | 7.6% | 4.2% | 7.1% |
| Total #ROUGE<0.2 | 250$^{\ddagger}$ | 360 | 323$^{\dagger}$ |

metrics over T5 with statistical significance. In conclusion, our human evaluation results indeed show that the Type+QA simulator improves T5 in terms of both generation relevance and naturalness.

                                                                    

***Manual analyses show that the reason for the improvements of Type+QA is indeed better answer typing.*** We conduct a similar manual analysis as Sec. 4 for Type+QA generations, shown in Table 7. We find 323 low-scoring examples from the Type+QA generations, which is significantly fewer than T5's 360. Each row of the table shows the percentages of low-scoring generations for one specific reason among all the low-scoring generations, for T5 with cooperativeness evaluation, T5, and Type+QA model. The first row of the table shows that Type+QA reduces the percentage of wrong answer types from 33.9% to 11.8% with statistical significance. This result shows that question answering knowledge and answer typing are useful for user simulation.

***Cooperativeness-aware evaluation reduces misevaluation.*** We show the cooperativeness-aware evaluation results for all the finetuned-based methods in Table 8 and Table 9 in the appendix. The numbers in the table are averaged, combining different cooperativeness. We see that all three finetuned models see significant improvements with cooperativeness-aware evaluation. The manual analysis in Table 7 also shows that cooperativeness mismatches drop significantly from 31.1% to only 2.4% when evaluation accounts for cooperativeness, and the total of low-scoring examples subsequently declines. These results jointly show that cooperativeness-aware evaluation can effectively reduce misevaluation.

***Open LLMs do not sufficiently understand the user simulation task.*** From Table 4 and Table 5, we see finetuning smaller generation models, in general, can achieve higher performance than the zero-shot open LLMs (i.e., Llama2 and Flan). When we dive into their generations, we see two main reasons for their failures:

(1) Llama2 struggles to understand user simulation tasks, instead, it often generates relevant but out-of-role information although prompted with persona instructions. Therefore, their scores are low in all of the columns.

> $i$ = "How is workplace diversity achieved and managed?"
> $q$ = "Diversity"
> $cq$ = "Are you looking for a definition of diversity?"
> $H$ = "No I am looking to achieve and manage it in the workplace."
> $LlaMa2$ = "Diversity means differences between people."

(2) Flan-xxl generations are too short, such as 'Yes.' or 'No.'. Evidence is that human generations have 8.05 words on average, while Flan-xxl only has 1.47 words. Such answers do not provide enough information for document retrieval. Although Flan-xxl achieves sufficiently high F1 in answer typing, which is even higher than finetuned T5, it gets extremely low generation scores and document retrieval scores.

In conclusion, all these open zero-shot LLMs cannot meet our expectations for user simulation. Our observations and conclusions about LLMs are aligned with recent findings about using LLMs for generation and simulation tasks [6, 59]. As a result, our work on better training user simulators is timely.

***Commercial LLMs are good at answering clarifying questions, but are currently not human-like in their responses.*** From Table 4 and Table 5, the main difference between GPTs and other LLMs is that GPTs score higher on all metrics, particularly on document retrieval metrics. While we have expected this result,

we are interested in why. Therefore, we randomly sampled 100 generated responses from GPT-4 and found that GPT-4 generations are highly templated and highly cooperative. It never simply answers a clarifying question as 'yes' or 'no' without explaining. Even humans are not as cooperative and patient, which is why GPT-4 is high on answer type F1 and document retrieval performance but low on the generation metrics. For example:

> $i$ = "Find the sports section of the Milwaukee Journal Sentinel."
> $q$ = "Milwaukee Journal Sentinel."
> $cq$ = "Which medium do you prefer the newspaper to be in?"
> $H$ = "I don't know"
> $GPT$ = "Either a physical or digital format works for me. "

Here, GPT-4 generates a good answer even when the human fails to give any useful information. While GPT-4 has the potential to be a perfect user simulator, we argue that it is inadequate since it does not behave as a real human. Human users could be as ambiguous in their responses as there were in the initial query; they may give short and incomplete answers or fail to understand the clarifying question. All of these behaviors should be expected by conversational search systems to generate clear questions that are easy to answer. Therefore, unlike GPT-4, a good user simulator should be able to simulate these human imperfections. This argument does not mean that models like GPT-4 should be avoided. Instead, future user simulation systems should view the task as more than just question answering or sequence-to-sequence text generation. Specifically, a good user simulator should both have good generation capability and be able to simulate human imperfections, which involves perhaps another layer of latent variables.

## 8 Conclusion

In this paper, we study the task of simulating user responses for clarifying questions in conversational search and provide insights into the task, focusing on the challenges with existing models and how the challenges may be addressed and identifying what is left to be solved. We find that finetuned T5 can significantly outperform existing user simulation systems. Rather than reporting it as the new state-of-the-art, we cast the question of what can be learned about user simulation. As part of the answering process, we conduct an in-depth manual analysis of the low-scoring generations of T5. It shows that aside from data noise, 38% of the generations are bad because T5 cannot effectively learn to generate responses of the correct answer type, and at least 45% of the 'bad' generations are due to misevaluations. We then propose a simple two-step model to correct the wrong answer types in generations, which is shown to reduce the above answer type error significantly from 38% to 12%. Further, we propose a data partition heuristic to account for an essential variable for user simulation, the *cooperativeness*, which substantially improves upon the existing evaluation standard. In the end, we compare our models with existing baselines and large language models. We show that our proposed system is the best and that existing large language models are inadequate for simulating users for conversational search. As a result, our investigation and work on better training user simulation models is timely.

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

# A  Appendices

## A.1  Answer Typing Algorithm

---

**Algorithm 1** Typing($cq, S_{idk}, W_{ne}$)

---

**Input:** A clarifying question $cq$, a list $S_{idk}$ of sentences expressing uncertainty or irrelevancy, and a list $W_{ne}$ of negation words.
**Output:** Type of $cq$
**if** $cq \in S_{idk}$ **then**
    **return** 'irrelevant'
**else if** 'yes' $\in cq[:3]$ **then**
    **return** 'yes'
**else if** Any([w $\in cq[:3]$ for w in $W_{ne}$]) **then**
    **return** 'no'
**end if**
**return** 'open'

---

## A.2  GPT-4 Instruction

> **role**: system
> **content**: In this task, imagine a user who wants to find information online and unintentionally asks a search system an ambiguous search query. To better understand their search intention, the system asks a clarifying question. Your goal is to generate answers to this clarifying question, based on the user's original search intent.

> **role**: user
> **content**: My search intent is: " + {query} + {intent} + "The system clarifying question is: " + {cq} + "How should I respond?

## A.3  Full Cooperativeness-aware Full Evaluation Results

**Table 8: Adding cooperativeness to all the finetuned models improve generation metrics on Qulac dataset. † indicate $p < 0.01$ statistical significance of improvements over the cooperative-unaware version using permutation test [14].**

| Model | Type | Generation Metrics | | | | Document Retrieval | | | | |
|---|---|---|---|---|---|---|---|---|---|---|
| | F1 | BLEU-3 | BLEU-4 | ROUGE-L | METEOR | nDCG1 | nDCG5 | nDCG20 | P@1 | MRR |
| T5-small | 34.1 | 23.7 | 19.0 | 40.8 | 43.2 | **0.215** | 0.209 | 0.188 | **0.279** | 0.388 |
| +Cooperativeness | 41.5$^\dagger$ | 27.8$^\dagger$ | 22.1$^\dagger$ | 47.7$^\dagger$ | 45.4$^\dagger$ | 0.206 | 0.203 | 0.182 | 0.268 | 0.377 |
| UnifiedQA | 41.7 | 23.3 | 18.7 | 40.9 | 43.4 | **0.215** | **0.210** | 0.188 | **0.279** | **0.390** |
| +Cooperativeness | 43.3$^\dagger$ | 28.3$^\dagger$ | 22.7$^\dagger$ | 47.8$^\dagger$ | 45.2$^\dagger$ | 0.203 | 0.205 | 0.183 | 0.263 | 0.376 |
| Type+UQA | 43.3 | 24.4 | 19.6 | 41.6 | 43.5 | 0.214 | **0.210** | **0.189** | 0.277 | **0.390** |
| +Cooperativeness | **50.1**$^\dagger$ | **30.1**$^\dagger$ | **24.3**$^\dagger$ | **50.5**$^\dagger$ | **46.7**$^\dagger$ | 0.203 | 0.202 | 0.182 | 0.265 | 0.376 |

**Table 9: Adding cooperativeness to all the finetuned models improve generation metrics on ClariQ dataset. † indicate $p < 0.01$ statistical significance of improvements over the cooperative-unaware version using permutation test [14].**

| Model | Type | Generation Metrics | | | | Document Retrieval | | | | |
|---|---|---|---|---|---|---|---|---|---|---|
| | F1 | BLEU-3 | BLEU-4 | ROUGE-L | METEOR | nDCG1 | nDCG5 | nDCG20 | P@1 | MRR |
| T5-small | 36.6 | 24.3 | 19.5 | 41.0 | 43.3 | **0.150** | 0.134 | 0.118 | **0.176** | **0.249** |
| +Cooperativeness | 41.8$^\dagger$ | 29.1$^\dagger$ | 23.4$^\dagger$ | 48.1$^\dagger$ | 45.4$^\dagger$ | 0.147 | 0.133 | 0.117 | 0.173 | 0.246 |
| UnifiedQA | 45.9 | 24.3 | 19.4 | 41.6 | 43.6 | 0.148 | 0.135 | **0.119** | 0.170 | 0.247 |
| +Cooperativeness | 48.1$^\dagger$ | 29.4$^\dagger$ | 23.5$^\dagger$ | 49.4$^\dagger$ | **46.7**$^\dagger$ | 0.146 | 0.134 | 0.117 | 0.169 | 0.245 |
| Type+UQA | 46.3 | 25.2 | 20.2 | 42.1 | 43.1 | 0.149 | **0.136** | **0.119** | 0.172 | **0.249** |
| +Cooperativeness | **53.4**$^\dagger$ | **30.4**$^\dagger$ | **24.3**$^\dagger$ | **50.0**$^\dagger$ | 45.6$^\dagger$ | 0.141 | 0.133 | 0.116 | 0.163 | 0.242 |

