# OpenReview forum: "An In-depth Investigation of User Response Simulation for Conversational Search"
_ACM.org/TheWebConf/2024/Conference — TheWebConf24 Oral_

### Official Review · Reviewer_d39p · 2023-11-21

**Novelty:** 4
**Technical Quality:** 4

**Review:**

In this work, the authors improved user simulation systems for conversational search by optimizing a smaller, fine-tuned language generation model. They tackled key challenges such as learning blind spots and evaluation methods, leading to major improvements over current systems and LLMs like GPT-4, offering practical insights for future advancements.

Below are some suggested directions for revision:
- It appears that line 231 is incomplete.
- The argument in lines 894-895 is interesting. Why do we need a user simulator that mimics imperfect human behavior? What are the advantages of such a simulator in practical applications? If the simulator produces vague or incomplete responses, how can we ensure that this doesn’t cause unnecessary confusion or frustration for the users?

**Questions:**

Regarding Table 6, the inferior performance of GPT-3.5 or GPT-4 in the generation task might be attributed to the use of automatic evaluation metrics. These models could generate responses with wording vastly different from the ground truth, potentially leading to lower evaluation scores. Is it possible that their actual performance is better than indicated?

**Reviewer Confidence:**

3: The reviewer is confident but not certain that the evaluation is correct

**Scope:**

3: The work is somewhat relevant to the Web and to the track, and is of narrow interest to a sub-community

---

### Official Review · Reviewer_foUc · 2023-11-23

**Novelty:** 4
**Technical Quality:** 4

**Review:**

This paper works on simulating user responses for clarifying questions in the scenario of information-seeking. This paper first did a systematic error analysis of a simulator based on a finetuned T5 model, showing it is necessary to consider either different answer types or cooperativeness levels for methodology. Then, the paper proposes an answer-type-driven user simulation and injects cooperativeness levels during training and evaluation; also, the paper considers either open-sourced or commercial large language models (LLMs) in a zero-shot setting as baselines. Experimental results show the effectiveness of the consideration of answer types and cooperativeness levels.

Overall, the paper is well-written and well-structured. The following are advantages:
1) a systematic error analysis of a simulator based on a finetuned T5 model, which is really valuable and gives the community a lot of insights,
2) propose to explicitly predict answer types and explicitly consider the cooperativeness levels during training and evaluation, which is beneficial for the final performance, and
3) the code is open-sourced.

However, the paper also has a lot of drawbacks.
1. LLM-based baselines are not used in an effective way. LLM-based baselines are all in a zero-shot setting in this paper. However, the literature has already shown that in-context learning (injecting some examples into LLMs) can improve the performance of LLMs a lot; also in-context learning has been widely used to improve the performance of LLMs in the literature. So it is not sure if the proposed model can still perform better than in-context learning-based LLMs.
2. A lot of important experimental details are unclear. 1) the proposed model and some baselines need fine-tuning; how many epochs do these models need? And how did the authors select the epoch for each model? Also, to consider the cooperativeness-awareness evaluation, the authors claim that they train two models on each data partition and evaluate the trained models on each partition separately. What is the size of each data subset? How does the model perform on each data partition?
3. In Tables 4 and 5, lack the significance tests between the proposed model and the strongest model (QA-Enhanced). The authors only conduct the significance tests between the proposed model and T5-small, which is weaker than QA-Enhanced. So we do not know if explicitly predicting answer types leads to significant improvement or not.
4. The simulation scenario in this paper is limited to answering the clarifying questions posted by systems, which is a specific and limited scenario in user simulation.

**Questions:**

1. What exactly is the difference between "no" and "irrelevant"?
2. The authors use the permutation test for significance tests. Why the authors do not consider more widely-used t-test?
3. To consider the cooperativeness-awareness evaluation, the authors claim that they train two models on each data partition and evaluate the trained models on each partition separately. What is the size of each data subset? How does the model perform on each data partition?

**Reviewer Confidence:**

4: The reviewer is certain that the evaluation is correct and very familiar with the relevant literature

**Scope:**

4: The work is relevant to the Web and to the track, and is of broad interest to the community

---

### Official Review · Reviewer_cMwT · 2023-11-24

**Novelty:** 4
**Technical Quality:** 4

**Review:**

Summary:

This paper addresses the challenge of simulating user responses in conversational search systems. The authors focus on improving user simulation systems, which are crucial for training and evaluating conversational search systems without requiring expensive human involvement. They identify key challenges in current simulation systems and present an improved model that includes a fine-tuned T5 model and a QA-enhanced user simulation system, addressing specific challenges like training blind spots.

-----------------
Strengths:
1. The problem is very important, and a well-executed user simulation in conversational search could potentially have a significant impact on many other IR and NLP tasks.
2. The paper leverages a high level of novelty by introducing an advanced approach to user response simulation, integrating QA knowledge and answer type prediction to enhance simulation quality.
3. The authors provide a thorough analysis of the T5 model's limitations in simulating user responses, identifying specific types of failures and their impact on performance.

-----------------
Weaknesses and comments:
1. The proposed solutions, while effective, involve complex models and methodologies that might pose challenges in terms of implementation and computational resources. In particular, since the methodology is not integrated in an end-to-end manner, there might be noise injected from different components of the system.
2. The potential biases introduced by such a methodology for user simulation should be discussed as limitations of the work.
3. Sections from line 96 to 124 are challenging to follow. It is suggested that the authors clearly name their contributions at the end of the introduction section for better clarity.
4. Some related works on clarifying questions are missing, and the authors could consider including them in their references.

https://dl.acm.org/doi/abs/10.1145/3524110

https://arxiv.org/pdf/2205.13771

https://arxiv.org/abs/2208.04882

5. It would be beneficial if the authors expanded their analysis beyond a single metric to identify challenging examples. They could consider examples where satisfaction scores differ across multiple metrics, not just ROUGE, especially metrics that go beyond lexical matching. This would help reduce Type 2 errors.
6. In Table 3, it is suggested that the authors separate the last row to avoid column name mismatches.

**Questions:**

1. Can you please clarify the differences between Type 2 and Type 3 errors? Type 3 appears to be a specific example of Type 2.
2. Could you provide some examples of Type 6 errors? It is recommended that the authors include more concrete examples of different types of errors in the appendices.
3. How much does the performance of Roberta in Section 5.3 impact the entire pipeline?
4. In Table 4 versus Table 5, there is a noticeable difference in the retrieval evaluation scores, while the generative evaluation scores are quite close. Do you have any insights into the reasons behind this discrepancy?

**Reviewer Confidence:**

3: The reviewer is confident but not certain that the evaluation is correct

**Scope:**

4: The work is relevant to the Web and to the track, and is of broad interest to the community

---

### Official Review · Reviewer_dDz1 · 2023-11-26

**Novelty:** 3
**Technical Quality:** 5

**Review:**

This paper proposes a user response simulator for conversational search. The simulator is built on a T5 model with QA-enhanced pre-training and answer-type classification strategy. This simple yet effective pipeline improves the final performance dramatically. This paper analyzes the challenges in existing conversational search methods. Specifically, many problems of T5 generation have been investigated in-depth, which is very inspiring. The paper is well-written and easy to understand.

Strengths:
+ The paper is well-written and easy to understand.
+ The analysis is very comprehensive, which is inspiring for the model design.
+ The experimental results demonstrate the effectiveness of the proposed method.

Weaknesses:
- This paper seems more like a technical report rather than a research paper. The method looks like a “plug-in” for the existing model (T5).
- Only a small model is used in experiments. I wonder if the method can get better results when T5-large/XL/XXL is used.
- Some descriptions are not clear. For example, what is the QA-Enhanced baseline?
- I think it is unfair to use zero-shot settings for LLMs, because the T5-model is fine-tuned for the task. Few-shot settings may be more suitable for experiments.

**Questions:**

See weaknesses.

**Reviewer Confidence:**

3: The reviewer is confident but not certain that the evaluation is correct

**Scope:**

4: The work is relevant to the Web and to the track, and is of broad interest to the community

---

### Official Review · Reviewer_JHKQ · 2023-11-29

**Novelty:** 5
**Technical Quality:** 5

**Review:**

This paper investigate the response errors of the generative user simulators in conversational search, and categorized them into 6 types, and proposed several quick fixes to different error types, extensive experiments show that effectiveness of the simple fixes. The findings justified through error analysis are inspiring and could be inspiring to the community.

Strengths:
1. Insights from error analysis could be inspiring to the the community.
2. Extensive experiments show that the proposed quick solutions to the issues identified in error analsys are simple but effective.

Weaknesses:
1. Prompting is tricky for baseline T5-small, some changes could potentials change the conclusion.
2. Would QA-enhanced pre-training introduce more type 5 errors, i.e., instead of generating answer to the clarification question, it might generate the answer to the query intent directly, right?
3. Compared to the big improvements on answer type prediction and cooperativeness-aware evaluation, the search performance improvement seems marginal, some more investigation or explanation would be interesting too.

**Questions:**

Would QA-enhanced pre-training introduce more type 5 errors, i.e., instead of generating answer to the clarification question, it might generate the answer to the query intent directly?

**Reviewer Confidence:**

3: The reviewer is confident but not certain that the evaluation is correct

**Scope:**

3: The work is somewhat relevant to the Web and to the track, and is of narrow interest to a sub-community

---

### Decision · Program_Chairs · 2024-01-22

**Decision:**

Accept (Oral)

**Comment:**

This paper proposes to leverage a small language model to improve user simulation systems for conversational seach.

 The paper was reviewed by five reviewers. The paper has clearly some merits.
 The main criticism is related to the competitiveness of the large language model adopted in the analysis. Please clarify this point in the camera-ready copy.